# Polynomial-Time in PDDL Input Size: Making the Delete Relaxation Feasible for Lifted Planning

**Pascal Lauer,**[1] **Álvaro Torralba,**[2] **Daniel Fišer,**[1,3] **Daniel Höller,**[1] **Julia Wichlacz,**[1] **Jörg Hoffmann**[1]

[1]Saarland University, Saarland Informatics Campus, Germany
[2]Aalborg University, Denmark
[3]Czech Technical University in Prague, Czech Republic
s8palaue@stud.uni-saarland.de, alto@cs.aau.dk, danfis@danfis.cz, {wichlacz,hoeller,hoffmann}@cs.uni-saarland.de

## Abstract

Polynomial-time heuristic functions for planning are commonplace since 20 years. But polynomial-time in which input? Almost all existing approaches are based on a grounded task representation, not on the actual PDDL input which is exponentially smaller. This limits practical applicability to cases where the grounded representation is "small enough". Previous attempts to tackle this problem for the delete relaxation leveraged symmetries to reduce the blow-up. Here we take a more radical approach, applying an additional relaxation to obtain a heuristic function that runs in time polynomial in the size of the PDDL input. Our relaxation splits the predicates into smaller predicates of fixed arity $K$. We show that computing a relaxed plan is still **NP**-hard (in PDDL input size) for $K \geq 2$, but is polynomial-time for $K = 1$. We implement a heuristic function for $K = 1$ and show that it can improve the state of the art on benchmarks whose grounded representation is large.

## Introduction

Heuristic search is a dominant paradigm for effective planning (e. g. (Hoffmann and Nebel 2001; Helmert and Domshlak 2009; Richter and Westphal 2010; Seipp 2019)). Polynomial-time computable heuristic functions are an essential ingredient to this success, and have been extensively investigated since 20 years. A particularly important technique is the delete relaxation (Bonet and Geffner 2001), which ignores negative effects (in a propositional encoding), essentially pretending that state variables accumulate their values rather than switching between them. Most state-of-the-art satisficing planning systems (which do not prove optimality of the solution returned) still use the delete relaxation or extensions thereof today (e. g. (Helmert et al. 2011; Keyder, Hoffmann, and Haslum 2014; Domshlak, Hoffmann, and Katz 2015; Cenamor, de la Rosa, and Fernández 2016)).

Virtually all of these approaches however suffer from the fact that "polynomial-time" is relative to the size of a grounded task representation. This is in contrast to the actual PDDL input of the planning system, which is *lifted*, specifying predicates and action schemas parameterized with variables ranging over a finite universe of objects. The grounded representation size is exponential in the size of that input, specifically in the arity of the predicates and action schemas.

This is not a practical problem when the grounded representation is small enough to be feasible. Yet in a variety of application scenarios that is not so (e. g. (Hoffmann et al. 2006; Koller and Hoffmann 2010; Koller and Petrick 2011; Haslum 2011; Matloob and Soutchanski 2016)).

Lifted planning has always been considered (e. g. (Penberthy and Weld 1992; Younes and Simmons 2003)), and indeed was dominant in the early 90s (Russell and Norvig 1995). There has been little progress however on transferring the wealth of known heuristic functions to the lifted setting. The only previous attempt considered the delete relaxation and leveraged symmetries to reduce the grounding blow-up in relaxed planning (Ridder and Fox 2014). Later works devised lifted domain analyses to reduce task size (Röger, Sievers, and Katz 2018; Sievers et al. 2019; Fišer 2020).

Here we take a more radical approach, applying an additional relaxation to obtain a heuristic that runs in time *polynomial in the size of the PDDL input*. Our relaxation splits the predicates $P(x_1, \ldots, x_n)$ in the PDDL input task $\Pi$ into smaller predicates $P_i(x_{i_1}, \ldots, x_{i_K})$ of arity $K$, where $\{i_1, \ldots, i_K\} \subseteq \{1, \ldots, n\}$ and $|\{i_1, \ldots, i_K\}| = K$. Specifically, every occurrence of $P$ is replaced by the conjunction of $P_i$ for all size-$K$ subsets of $P$'s parameters. The size of the resulting lifted planning task $\Pi|_K$ is exponential only in $K$, hence polynomial for fixed $K$. This is a relaxation in conjunction with the delete relaxation, in the sense that every plan for $\Pi$ is a delete-relaxed plan for $\Pi|_K$. We show that computing a delete-relaxed plan for $\Pi|_K$ is still **NP**-hard (in PDDL input size) for $K \geq 2$, but is polynomial-time for $K = 1$. We implement a heuristic function for $K = 1$, and we devise an optimization that leverages some $K = 2$ information from static predicates.

We implement our heuristic on top of the Power Lifted Planner recently introduced by Corrêa et al. (2020), which grounds predicates and actions lazily during the forward search process. Standard International Planning Competition (IPC) benchmarks are not suited for evaluation as they are designed to challenge search rather than the grounding process. The only benchmarks currently available to challenge grounding are the ones by Areces et al. (2014), which contain action schemas of large arity (their work was about splitting large action schemas into several smaller ones). Correa et al. used these benchmarks. Here we go beyond

this by exploring different reasons for being hard-to-ground: (a) large action-schema arity; (b) large predicate arity, which entails large action-schema arity but may have other consequences; (c) large object universe, which can be problematic even for small action/predicate arity. For (a) we use Areces et al.'s benchmarks; for (b) we generalize two IPC domains (Visitall and Childsnack) that have a naturally scalable dimensionality parameter; for (c) we generate larger instances of some IPC benchmark domains in a spirit similar to one experiment reported about by Ridder and Fox (2014). For both (b) and (c), we take care to generate huge instances that are however within (and just beyond) reach of current lifted planners, in a manner similar to typical benchmark design in the IPC (Long and Fox 2003; Hoffmann et al. 2006; Gerevini et al. 2009; Coles et al. 2012; Vallati, Chrpa, and McCluskey 2018; Torralba, Seipp, and Sievers 2021). The design of this benchmark suite tailored to the evaluation of lifted planning is another contribution of our work. Our experiments show that our new polynomial-time lifted heuristic functions can improve the state of the art on these benchmarks, in particular through combination with goal counting.

## Background

A lifted planning task is a tuple $\Pi = (\mathcal{P}, \mathcal{O}, \mathcal{A}, \mathcal{I}, \mathcal{G})$ where $\mathcal{P}$ is a set of (first-order) *predicates*, $\mathcal{A}$ is a set of *action schemas*, $\mathcal{O}$ is a set of *objects*, $\mathcal{I}$ is the *initial state*, and $\mathcal{G}$ is the *goal*. Predicates $P \in \mathcal{P}$ have a tuple of parameter variables $X_P$, and we write $P(x_1, \ldots, x_{|X_P|})$ whenever we want to explicitly declare them. The arity of $P$ is $|X_P|$. We denote individual parameters with $x, y, z \in X_P$. We can instantiate a predicate, i.e., replace the set of parameters by objects from $\mathcal{O}$ or other variables by applying a substitution. If all variables have been replaced by objects, then $P$ is a ground predicate or *atom*. The set of ground atoms of $P$, resulting from all possible substitutions of variables in $X_P$ by objects in $\mathcal{O}$, is denoted $P^{\mathcal{O}}$. By $\mathcal{P}^{\mathcal{O}}$ we denote the set of all ground atoms in the task. The initial state and goal are sets of ground atoms.

An action schema $a = (X_a, pre(a), add(a), del(a))$ is a tuple with a set of parameter variables $X_a$, as well as *preconditions*, *add list*, and *delete list*, all of which are sets of predicates in $\mathcal{P}$ instantiated by substituting each of their variables by some element in $X_a \cup \mathcal{O}$. As with predicates, the arity of $a$ is $|X_a|$, and we can instantiate action schemas by replacing each $x \in X_a$ by some $o \in \mathcal{O}$ to obtain ground actions. The set of ground actions (or actions for short) is $\mathcal{A}^{\mathcal{O}}$. Note that, as the arity of predicates and action schemas is not bounded, $\mathcal{P}^{\mathcal{O}}$ and $\mathcal{A}^{\mathcal{O}}$ are of size exponential in the size of $\Pi$.

A ground action $a$ is applicable in a state $s$ if $pre(a) \subseteq s$. The resulting state of applying $a$ on $s$ is $(s \setminus del(a)) \cup add(a)$. A sequence of actions $a_1, \ldots, a_n$ is applicable in a state $s$ if there exists a sequence of states $s_0, \ldots, s_n$ such that $s_0 = s$, and $s_i$ is the result of applying $a_i$ in $s_{i-1}$ for all $i \in [1, k]$. We deal with the problem of finding a plan for an arbitrary planning task $\Pi$, that is, a sequence of ground actions applicable in $\mathcal{I}$ and resulting in some $s_n$ such that $\mathcal{G} \subseteq s_n$.

The delete-relaxation consists of ignoring the delete list

$del(a)$ of all action schemas. The FF heuristic (Hoffmann and Nebel 2001) estimates the distance from any state $s$ as the length of a relaxed plan, which can be computed in polynomial time in the size of the ground task.

Previous work by Corrêa et al. (2020) has shown that evaluating whether there exists an instantiation of an action schema that is applicable on a state is closely connected to the problem of resolving conjunctive queries in database theory (Ullman 1989). A database $DB = (D, \mathcal{R})$ has a domain $D$ and a set of relations $\mathcal{R}$ over $D$, such that each $R_i \in \mathcal{R}$ is a set $R_i \subseteq D^{ar(R_i)}$ where $ar(R_i)$ is the arity of $R_i$. Following planning nomenclature, $s_{DB} = \bigcup_{R_i \in \mathcal{R}} R_i$ is a state over a set of predicates $\mathcal{P}_{DB} = \{P_i \mid R_i \in \mathcal{R}\}$ and objects $\mathcal{O}_{DB} = D$.

A conjunctive query $Q$ over a database $DB$ consists of a set of variables $X_Q$ and set of predicates $P_i \in \mathcal{R}$ instantiated with objects in $D$ and/or variables in $X_Q$. $Q$ corresponds to the problem of finding a substitution of variables in $X_Q$ by objects in $D$ such that all atoms in the query belong to the database. The preconditions of an action schema can be seen as a conjunctive query that corresponds to finding which instantiations of the action schema are applicable in $s_{DB}$. Evaluating conjunctive queries (hence, lifted successor generation) is **NP**-hard in general (Chandra and Merlin 1977), but it is tractable for acyclic conjunctive queries (Yannakakis 1981). We say that an action schema has acyclic preconditions if the corresponding conjunctive query is acyclic. For a detailed introduction, we refer the reader to the work by Corrêa et al. (2020).

As running example we will use an extension of the Visitall IPC domain, where an agent must visit all tiles in a 2D grid. We generalize this to $d$-dimensional hypercube grids with side length $l$, and we permit goals requiring to visit a subset of the locations. Figure 1 sketches the encoding of our running example for $d = 3$.

The positions in the hypercube are tuples of indices in $\mathbb{N}_l = \{i \in \mathbb{N} \mid 1 \leq i \leq l\}$. The set of all positions is $\mathbb{N}_l^d$. Similar to the original domain, the player is at some position in the beginning and can move to adjacent positions. Note that we specify a separate move- action schema for each dimension, so that we need to encode adjacency only over the numbers $\{1, \ldots, l\}$ (next predicate), not over positions (number tuples) as in the standard benchmark. Furthermore, instead of requiring the player to visit all positions, the requirement is to visit a subset of positions $G \subseteq \mathbb{N}_l^d$. This example's grounded representation is exponential in $d$ (which equals maximal predicate arity) as it needs to enumerate all possible positions so $|\mathcal{P}^{\mathcal{O}}| \geq l^d$. The same blowup occurs in the lifted task in case all positions need to be visited, i.e. if $\mathcal{G} = \mathbb{N}_l^d$. Yet if the number of goal positions is polynomial in $d$, then the ground task is exponentially larger than the lifted task.

## Complexity of Lifted Relaxed Planning

It is well known that a relaxed plan can be computed in polynomial time in the size of the ground task (Bonet and Geffner 2001; Hoffmann and Nebel 2001). In lifted planning, however, there are (at least) two sources of hardness:

$$\mathcal{P}=\{at(x,y,z), visited(x,y,z), next(x,x')\}$$
$$\mathcal{O}=\{1,\ldots,l\}$$
$$\mathcal{I}=\{at(1,1,1)\} \cup \{next(i,j) \mid 1 \le i,j \le l, |i-j|=1\}$$
$$\mathcal{G}=\{visited(3,2,4)\}$$
$$\mathcal{A}=\{\texttt{move-x}(x,y,z,x'), \texttt{move-y}(x,y,z,y'), \texttt{move-z}(x,y,z,z')\}$$
where $\texttt{move-x}(x,y,z,x')$:
$$pre : \{at(x,y,z), next(x,x')\}$$
$$add : \{at(x',y,z), visited(x',y,z)\}$$
$$del : \{at(x,y,z)\}$$

Figure 1: Running example: $d$-dimensional Visitall with $d = 3$.

$$\mathcal{P}=\{at_1(x), at_2(x), at_3(x), v_1(x), v_2(x), v_3(x), n_1(x), n_2(x)\}$$
$$\mathcal{O}=\{1,\ldots,l\}$$
$$\mathcal{I}=\{at_1(1), at_2(1), at_3(1)\} \cup \{n_1(i), n_2(j) \mid i,j \in [1,l]\}$$
$$\mathcal{G}=\{v_1(3), v_2(2), v_3(4)\}$$
$$\mathcal{A}=\{\texttt{move-x}(x,y,z,x'), \texttt{move-y}(x,y,z,y'), \texttt{move-z}(x,y,z,z')\}$$
where $\texttt{move-x}(x,y,z,x')$ :
$$pre : \{at_1(x), at_2(y), at_3(z), n_1(x), n_2(x')\}$$
$$add : \{at_1(x'), at_2(y), at_3(z), v_1(x'), v_2(y), v_3(z)\}$$
$$del : \{at_1(x), at_2(y), at_3(z)\}$$

Figure 2: 1-ary (also: unary) relaxation for our running example.

1. The number of ground actions $|\mathcal{A}^{\mathcal{O}}|$ is exponential in lifted task size. This might incur exponential effort in determining applicable actions, a key step underlying all known relaxed planning algorithms.

2. The number of ground atoms $|\mathcal{P}^{\mathcal{O}}|$ also is exponential in lifted task size. Hence both trivial upper bounds on relaxed plan length – number of ground atoms, number of ground actions – are not polynomial in this setting.

Indeed, delete-relaxed planning on lifted planning tasks was shown to be **EXPTIME**-complete (Erol, Nau, and Subrahmanian 1995). To better understand the sources of complexity at play here, we consider two further restrictions, and show that the problem is still hard (1) even if the predicate arity is restricted to be constant and (2) even if checking action applicability can be performed efficiently.

The first result follows directly from results of recent work on the problem of lifted successor generation (Corrêa et al. 2020), which showed an equivalence to answering conjunctive queries, viewing action-schema preconditions as queries over the state. Answering such a query is hard if it is cyclic in a certain sense. We can use this insight for a simple reduction from query answering to planning, in which a relaxed plan exists iff an applicable action exists in the initial state iff the answer to a query is true.

**Theorem 1.** *It is NP-hard to decide relaxed plan existence in lifted planning, even if predicate arity is constant.*

*Proof.* We use a reduction from conjunctive queries, which are **NP**-hard even with 2-arity predicates (Chandra and Merlin 1977). Let $Q$ be a conjunctive query over a database *DB*. Consider a task $\Pi_Q = (\mathcal{P}, \mathcal{O}, \mathcal{A}, \mathcal{I}, \mathcal{G})$ where $\mathcal{P} = \mathcal{P}_{DB} \cup \{goal\}$ (*goal* is a 0-arity predicate), $\mathcal{I} = s_{DB}$, $\mathcal{G} = \{goal\}$, and $\mathcal{A} = \{a\}$ with $X_a = X_Q$, $pre(a) = Q$, $add(a) = \{goal\}$. Then $\Pi_Q$ is (relaxed) solvable if and only if some instantiation of $a$ is applicable on $\mathcal{I}$: i.e., if the conjunctive query $Q$ is not empty. $\square$

For our second result, we encode a counter with $n$ binary variables, where the plan is to count from 0 to $2^n - 1$. Notably, this can be done with extremely simple action schemas, in particular ones with acyclic precondition queries, so that this source of complexity is independent from the previous one:

**Theorem 2.** *There exist families of lifted planning tasks $\{\Pi_1, \Pi_2, \ldots\}$ with acyclic action-schema preconditions where delete-relaxed plans have exponential length.*

*Proof.* We define $\Pi_n$ as $(\mathcal{P}_n, \mathcal{O}, \mathcal{A}_n, \mathcal{I}_n, \mathcal{G}_n)$ where $\mathcal{O} = \{o_0, o_1\}$, $\mathcal{P}_n = \{P(x_1, \ldots, x_n)\}$, $\mathcal{I}_n = \{P(o_0, \ldots, o_0)\}$, $\mathcal{G}_n = \{P(o_1, \ldots, o_1)\}$, and $\mathcal{A}_n = \{a_1, \ldots, a_n\}$. The action schemas are $a_i(x_1, \ldots, x_{i-1})$ for $1 \le i \le n$ (note that $a_1$ has no parameters), with $pre(a_i) = \{P(x_1, \ldots, x_{i-1}, o_0, o_1, \ldots, o_1)\}$ and $add(a_i) = \{P(x_1, \ldots, x_{i-1}, o_1, o_0, \ldots, o_0)\}$. Every relaxed plan has to achieve $2^n - 1$ ground atoms, applying $2^n - 1$ actions. $\square$

## K-ary Predicate Splitting

To simplify the computation of relaxed plans at a lifted level, we apply a relaxation based on splitting each $n$-ary predicate into several $K$-ary predicates where $K < n$ is a parameter for our approach. For a given $K$, the splitting operation ($|_K$) replaces the predicate by $\binom{n}{K}$ sub-predicates that correspond to all possible combinations of $K$ parameters. For example, consider the predicate $at(x, y, z)$ from our example in Figure 1. Then, $at|_1 = \{at_1(x), at_2(y), at_3(z)\}$ and $at|_2 = \{at_1(x,y), at_2(y,z), at_3(x,z)\}$. The same operation can be applied to ground atoms in the initial state or goal as well as to action schemas by applying it to *pre*, *add* and *del* (e.g. see Figure 2). We also define this operation over sets of predicates, action schemas, etc, as the union of applying ($|_K$) to each individual in the set, e.g., $\mathcal{P}|_K = \bigcup_{P \in \mathcal{P}} \{P|_K\}$.

Based on this splitting operation, we define the $K$-ary relaxation of a lifted planning task.

**Definition 1** (K-ary Relaxation). *Let $\Pi = (\mathcal{P}, \mathcal{O}, \mathcal{A}, \mathcal{I}, \mathcal{G})$ be a lifted planning task and $K$ be a constant. We define the $K$-ary relaxed task $\Pi|_K$ as a task $(\mathcal{P}|_K, \mathcal{O}, \mathcal{A}|_K, \mathcal{I}|_K, \mathcal{G}|_K)$.*

Obviously, plans for $\Pi|_K$ are not necessarily plans for $\Pi$, so this is an approximation. Observe that, together with the delete relaxation, it is an over-approximation and thus indeed constitutes a relaxation:

**Theorem 3.** *Let* $\Pi = (\mathcal{P}, \mathcal{O}, \mathcal{A}, \mathcal{I}, \mathcal{G})$ *be a lifted planning task, $K$ be a constant, and $\Pi|_K = (\mathcal{P}|_K, \mathcal{O}, \mathcal{A}|_K, \mathcal{I}|_K, \mathcal{G}|_K)$ be the $K$-ary relaxed task. Then every plan for $\Pi$ is a delete-relaxed plan for $\Pi|_K$.*

*Proof.* Every plan for $\Pi$ is a delete-relaxed plan for $\Pi$, so it suffices to show that delete-relaxed plans are preserved in $\Pi|_K$. Let $a_1, \ldots, a_n$ be a delete-relaxed plan for $\Pi$, let $\mathcal{I} = s_0, s_1, \ldots, s_n$ be the (relaxed) states traversed by that plan in $\Pi$, and let $\mathcal{I}|_K = s'_0, s'_1, \ldots, s'_n$ be the states traversed by that plan in $\Pi|_K$. We show, by induction over $i$, that (1) $a_i|_K$ is applicable in $s'_{i-1}$ and (2) $s'_i = s_i|_K$. For the base case $i = 0$, (1) is empty and (2) holds by construction. For the inductive case, say the claim holds for $i-1$. Then $s'_{i-1} = s_{i-1}|_K$, so (1) $a_i|_K$ is applicable in $s'_{i-1}$ by construction of $pre(a_i|_K)$. Regarding (2),

$$
\begin{aligned}
s_i|_K &= [s_{i-1} \cup add(a_i)]|_K && \text{[Def. of action application]} \\
&= [s_{i-1}]|_K \cup [add(a_i)]|_K && \text{[Prop. of set projection]} \\
&= s'_{i-1} \cup [add(a_i)]|_K && \text{[Induction Hypothesis]} \\
&= s'_{i-1} \cup add(a_i|_K) && \text{[Def. of } a|_K] \\
&= s'_i && \text{[Def. of action application]}
\end{aligned}
$$
$\square$

Importantly, the same is not true without the delete relaxation: we do not have a guarantee that every plan for $\Pi$ is a (non-delete-relaxed) plan for $\Pi|_K$. This is because, when deleting $P|_K$ in $\Pi|_K$, we may delete split atoms associated also with other instantiations of the same predicate. For example, in a state that contains both $P(a, b)$ and $P(a, c)$, an action that deletes $P(a, b)$ in $\Pi$ deletes $P_1(a)$ in $\Pi|_1$, so that the outcome state in $\Pi|_1$ does *not* contain $P(a, c)|_1$.[1]

Regarding the complexity of delete-relaxed planning in $\Pi|_K$, all predicates in $\Pi|_K$ have a bounded arity of at most $K$. So the length of a relaxed plan for $\Pi|_K$ is polynomial in the size of $\Pi$ and the complexity source identified by Theorem 2 disappears. The complexity source identified by Theorem 1 remains valid though for $K \geq 2$, as answering conjunctive queries is **NP**-hard even in this case. Indeed, the action schemas resulting from 2-ary predicate splitting have cyclic preconditions. So deciding whether a relaxed plan for $\Pi|_K$ exists remains hard in general. Here we exploit the case $K = 1$, *unary* predicate splitting, where as we shall see next relaxed plans can be computed in polynomial time.

## Unary Relaxed Planning

Even though the number of ground actions in the unary-split task $\Pi|_1$ is still exponential in the size of $\Pi$, delete-relaxed plans for $\Pi|_1$ can be computed in polynomial time.

The unary relaxation heuristic ($h^{\text{ur}}$, Alg. 1) accomplishes this, in a manner analogous to the computation of relaxed plans in ground tasks (Hoffmann and Nebel 2001). It constructs a best-supporter function that maps each ground atom in $\Pi|_1$ to a ground action. Starting at the initial state, the algorithm iteratively computes a larger set of reachable atoms

---

[1] Higher-arity predicates can be compiled into binary predicates equivalently, i. e., without information loss. This compilation however requires the introduction of an ID (a new object) for every ground atom, and is thus of size exponential in the lifted encoding.

---

**Algorithm 1:** Unary Relaxed Plan Computation ($h^{\text{ur}}$)

**Input:** Planning Task: $\Pi = (\mathcal{P}, \mathcal{O}, \mathcal{A}, \mathcal{I}, \mathcal{G})$
**Output:** Relaxed Plan for $\Pi|_1$ or "Unsolvable"

1   $F_0 \leftarrow \mathcal{I}|_1$
2   $i \leftarrow 0$
3   **do**
4      $i \leftarrow i + 1$
5      $F_i \leftarrow F_{i-1}$
6      **foreach** $P(o) \in \mathcal{P}|_1^{\mathcal{O}} \setminus F_{i-1}$ **do**
7          $bs[P(o)] \leftarrow$
         GetBestSupporter$(\Pi, P(o), F_{i-1})$
8          **if** $bs[P(o)] \neq None$ **then**
9             $F_i \leftarrow F_i \cup \{P(o)\}$

10   **while** $\mathcal{G}|_1 \nsubseteq F_i \wedge F_i \neq F_{i-1}$
11   **if** $\mathcal{G}|_1 \subseteq F_i$ **then**
12      **return** ExtractRelaxedPlan$(\Pi, bs)$
13   **else return** "Unsolvable"

14   **function** GetBestSupporter $(\Pi, P(o), F)$:
15      **foreach** $a(x_1, \ldots, x_n) \in \mathcal{A}|_1, i \in \{1, \ldots, n\}$, s.t. $P(x_i) \in add(a)$ **do**
         /* $\exists o_1, \ldots, o_n \in \mathcal{O}$ s.t.
         $o_i = o \wedge pre(a(o_1, \ldots, o_n)) \subseteq F$      */
16          **foreach** $j \in \{1, \ldots, n\}$ **do**
17             $\mathcal{O}_j \leftarrow \{o\}$ **if** $j = i$ **else** $\mathcal{O}$
18             **foreach** $Q(x_j) \in pre(a)$ **do**
19                 $\mathcal{O}_j \leftarrow \mathcal{O}_j \cap \{o' \mid Q(o') \in F\}$
20          **if** $\forall_{j \in [1,n]} \mathcal{O}_j \neq \emptyset$ **then**
21             **return** any $a(o_1, \ldots, o_n)$ s.t. $o_j \in \mathcal{O}_j$
22      **return** None

23   **function** ExtractRelaxedPlan $(\Pi, bs)$:
24      $queue \leftarrow \mathcal{G}|_1 \setminus \mathcal{I}|_1$
25      $plan \leftarrow \langle \rangle$
26      **while** $queue \neq \emptyset$ **do**
27          $f \leftarrow queue.\text{pop}()$
28          **if** $bs[f] \notin plan$ **then**
29             $plan.\text{append}(bs[f])$
30             $queue \leftarrow queue \cup pre(bs[f]) \setminus \mathcal{I}|_1$
31      **return** reverse(plan)

---

$F_i$, enabling in each iteration the preconditions of best supporter actions for new atoms. All atoms in the $i$th layer, $F_i$, are reachable by applying an action whose preconditions have been reached in the previous layers. In other words, among all possible supporters, we choose one whose precondition has minimal $h^{max}$ value in $\Pi|_1$ (Bonet and Geffner 2001).

The key to polynomial-time behavior is that, in contrast to the algorithms commonly used on ground tasks, we do not enumerate applicable ground actions in each step. Instead, we merely keep track of a best supporter for each ground atom. There are polynomially many ground atoms in $\Pi|_1$, and it turns out we can identify best supporters efficiently. Namely, the function GetBestSupporter iterates over

action schemas instead of ground actions. We consider action schemas $a(x_1, \ldots, x_n) \in \mathcal{A}$ where $P(x_i) \in add(a)$. Then, we check for each parameter $j \neq i$ *separately* with which objects can $x_j$ be instantiated such that the action preconditions are contained in $F$. The check in line 20 evaluates to true iff $\exists o_1, \ldots, o_{j-1}, o_{j+1}, \ldots, o_n \in \mathcal{O}$ such that $pre(a(o_1, \ldots, o_{i-1}, o, o_{i+1}, \ldots, o_n)) \subseteq F$, i.e., iff there exists an instantiation of $a$ that achieves the atom $P(o)$ and whose precondition is contained in $F$. This holds because, all preconditions being unary, the objects able to instantiate each parameter can be checked independently.

Once the best supporters have been chosen, relaxed plan extraction (`ExtractRelaxedPlan`) can easily be done in polynomial time. We process atoms one by one, starting with the goal atoms, inserting best-supporter actions into the relaxed plan and adding their preconditions to the atoms queue.

**Theorem 4.** *Algorithm 1 runs in time polynomial in the size of $\Pi$, and returns a delete-relaxed plan for $\Pi|_1$ iff such a plan exists.*

*Proof.* If Algorithm 1 returns a plan, then it is a valid relaxed plan for $\Pi|_1$. This plan achieves all goals, as it contains $bs[f]$ for all $f \in \mathcal{G}|_1$, and $f \in add(bs[f])$. It is applicable in $\mathcal{I}$ as, for every action included in the plan, a supporting action will be inserted for every precondition not already true in $\mathcal{I}$.

If the algorithm terminates without finding a relaxed plan (i.e., the main loop ends due to $F_i = F_{i-1}$) then the last $F_i$ contains all reachable ground atoms, so some goal atom is unreachable from $\mathcal{I}$, meaning that no plan exists.

The algorithm runs in time polynomial in the size of $\Pi$. Those "for each" loops in Algorithm 1 that iterate over elements of the lifted task $\Pi$ obviously perform a polynomial number of iterations. The same is true for all other loops because the number of ground atoms $|\mathcal{P}|_1^{\mathcal{O}}|$ is polynomial in $|\Pi|$. This is immediate for the loop in line 6, as it simply iterates over the elements in $\mathcal{P}|_1^{\mathcal{O}}$. Similarly, the main do-while loop in line 10 has at most one iteration per ground atom because in each iteration at least one new ground atom is added to $F_i$ (or else the loop stops immediately). Regarding the relaxed plan extraction, the loop in line 26 iterates over the elements in the goal, $\mathcal{G}|_1$ and the preconditions of all selected best supporters. Note that there is at most one supporter for each ground atom in $\mathcal{P}|_1^{\mathcal{O}}$, so at most polynomially many atoms are inserted in the queue. $\square$

Our implementation of Algorithm 1 has an additional tie-breaking for the choice in line 21, selecting the object in $\mathcal{O}_j$ whose preconditions $Q(o)$ for all $Q(X_j) \in pre(a)$ where inserted first (i.e., achieved in an earlier layer). The intuition is that the preconditions of those best supporters are easier to achieve from the initial state, leading to better relaxed plans. Note that, as in the FF heuristic, the relaxed plans are not guaranteed to be optimal, therefore $h^{\mathrm{ur}}$ is not an admissible heuristic.

In our running example, all (unary) atoms are reachable in layer $F_1$. The resulting relaxed plan is $\texttt{move-x}(1, 1, 1, 3), \texttt{move-y}(1, 1, 1, 2), \texttt{move-z}(1, 1, 1, 4)$, so

that $h^{\mathrm{ur}}(\mathcal{I}) = 3$. Note that this is a valid relaxed plan for the unary task from Figure 2.

## Disambiguation with Static Predicates

We next devise an optimization leveraging static predicates to obtain a better heuristic function. To motivate this, consider again the running example in Figures 1 and 2. Under unary relaxation, the heuristic value is at most $3 \cdot |\mathcal{G}|_1$, regardless of which positions need to be visited in the goal, because we can move from any coordinate to any other coordinate (e.g., $\texttt{move-x}(1, 1, 1, 3)$ is applicable in the initial state, going from $x$-coordinate 1 to 3 in a single step). This happens because we split not only the $at$ predicate, but also the $next$ predicate used to determine which numbers are adjacent to each other.

Ideally, we would like to at least obtain something resembling Manhattan distance, still separating the dimensions (by splitting the $at$ and $visited$ predicates), but capturing movements within each dimension correctly. To achieve the latter, we must preserve the adjacency information in $next$. It turns out that this is indeed possible while still keeping the computational cost at bay, i.e., while preserving independence across the parameters of each action schema.

We modify the `GetBestSupporter` function in Algorithm 1, through a refined version of object collection at each position $j$ in the second foreach loop. Say we need to support the atom $P(o)$, with action schema $a(x_1, \ldots, x_n)$ and $i \in \{1, \ldots, n\}$ such that $P(x_i) \in add(a)$. Our modification replaces the full set of objects $\mathcal{O}$ assigned to $\mathcal{O}_j$ in line 17 by a more restricted set $\mathcal{O}_{a(x_i=o), j}$. That set contains only those objects which, when $x_i$ is instantiated with $o$, can instantiate $x_j$ while satisfying the static predicates. Precisely, let $\mathcal{P}_{st}$ be the set of static predicates, i.e., $P_{st} \in \mathcal{P}$ such that $P_{st} \notin add(a)$ for any $a \in \mathcal{A}$. For any $a \in \mathcal{A}$ and $x_i, x_j \in X_a$, we denote the set of static preconditions of $a$ and pairs of subindices that correspond to $x_i$ and $x_j$ in the precondition by $pre_{st}(a, x_i, x_j)$. Formally, $pre_{st}(a, x_i, x_j) = \{\langle P_{st}, k, l \rangle \mid P_{st} \in \mathcal{P}_{st}, P_{st}(x'_1, \ldots, x'_m) \in pre(a), x'_k = x_i, x'_l = x_j\}$. In our example, $pre_{st}(\texttt{move-x}, x, x') = \{\langle next, 1, 2 \rangle\}$ as there is a precondition with the static predicate $next$ having $x$ as first and $x'$ as second argument. Then, $\mathcal{O}_{a(x_i=o), j} := \bigcap_{\langle P_{st}, k, l \rangle \in pre_{st}(a, x_i, x_j)} \{o' \in \mathcal{O} \mid \exists o_1, \ldots, o_m \text{ s.t. } P_{st}(o_1, \ldots, o_m) \in \mathcal{I}, o_k = o, o_l = o'\}$. We denote the resulting heuristic function with $h^{\mathrm{ur\text{-}d}}$.

For example, say we need to achieve $at_1(3)$, and consider $\texttt{move-x}(x, y, z, x')$ with $x' = 3$. In the previous version of Algorithm 1, the set of objects associated with the first argument $j = 1$ will be simply $\mathcal{O}$, allowing to move to 3 from anywhere. In our refined algorithm, that object set is $\{2, 4\}$ due to the static precondition $next(x, x')$. The relaxed plan for our running example then is $\texttt{move-x}(1, 1, 1, 2), \texttt{move-x}(2, 1, 1, 3), \texttt{move-y}(1, 1, 1, 2), \texttt{move-z}(1, 1, 1, 2), \texttt{move-z}(1, 1, 2, 3), \texttt{move-z}(1, 1, 3, 4)$, resulting in heuristic value $h^{\mathrm{ur\text{-}d}}(\mathcal{I}) = 6$.

Note that this is only a (tractable) approximation of the set of instantiations valid according to the static predicates when using predicate splitting with $K = 2$ for static predicates and $K = 1$ for the rest. We are instantiating each parame-

ter independently, and therefore the set of objects associated with each parameter can be computed in polynomial time, at expenses of admitting instantiations that would not satisfy the static preconditions in the original problem or even within the $K = 2$ relaxation. Note further that one could apply this disambiguation to non-static predicates as well. But that would require to re-compute the set of objects, not only for every state during search, but also at each iteration of the algorithm, for each layer $F_i$. Restricting the disambiguation to static predicates, in contrast, allows us to pre-compute the sets of objects for each action schema, object, and parameter position once before the search starts, with respect to $\mathcal{I}$ instead of $F$.

## Experiments

We implemented $h^{\text{ur}}$ and the static disambiguation variant $h^{\text{ur-d}}$ on top of the Power Lifted (PWL) planner (Corrêa et al. 2020), which uses Breadth First Search (BFS) and Greedy Best-First Search (GBFS) with goal counting ($h^{\text{gc}}$) (Fikes and Nilsson 1971). Apart from GBFS with $h^{\text{ur}}/h^{\text{ur-d}}$, we also consider a combination with goal counting, using our heuristic for tie-breaking (Röger and Helmert 2010) among nodes with the same $h^{\text{gc}}$ value. We also compare against the other existing lifted heuristic search planner, L-RPG (Ridder and Fox 2014). For additional reference, we report results for grounded planning, running Fast Downward's (FD) (Helmert 2006) GBFS with the $h^{\text{gc}}$ and $h^{\text{FF}}$ (Hoffmann 2001) heuristics. In all runs of PWL the successor generator based on Yannakakis' algorithm (Yannakakis 1981) was selected. The experiments were run on a cluster of machines with Intel Xeon E5-2650 CPUs with a clock speed of 2.30GHz using the Lab framework (Seipp et al. 2017). Timeout and memory limits were set to 30 minutes and 4GB respectively for all runs. All source code, experimental results and benchmarks are publicly available (Lauer et al. 2021).

### Benchmark Design

We contribute a new benchmark set for lifted planning, exploring different reasons why a planning task may be hard to ground: (a) large action-schema arity; (b) large predicate arity, which entails large action-schema arity but may have other consequences; and (c) large object universe, which can be problematic even for small arity domains.

Our benchmarks of category (a) are simply the ones previously used to evaluate hard-to-ground planning (Areces et al. 2014; Corrêa et al. 2020). These consist of three domains: Genome Edit Distance (GED) (Haslum 2011), Organic Synthesis (Masoumi, Antoniazzi, and Soutchanski 2015) and Pipesworld-Tankage (Hoffmann et al. 2006). We include GED here for historical reasons only: the encodings supported by the PWL planner are actually not that hard to ground, and Fast Downward's pre-process succeeds on all its instances.[2]

We design new benchmarks for categories (b) and (c). We extend standard IPC domains, aiming for large instances

---

[2]Other encodings of GED are harder to ground, but they use advanced PDDL features unsupported by PWL and our planner.

that are hard to ground, but with simple enough goals such that some instances can be solved by current lifted planners. We scale the instances by parameters controlling task size and goal complexity, allowing us to observe how the performance of different planners is affected.

For (b), we create new variants of Visitall and Childsnack, which have a naturally scalable dimensionality parameter that controls predicate arity. The Visitall extension is our running example. We create instances with $d \in \{3, 4, 5\}$ dimensions. For each of these cases we control the difficulty of instances by changing the number of goal locations from 1 to 3 and their relative position with respect to the starting location, close or far. For each of these categories we create 10 instances by scaling the size of the hypercube, starting at $l = 6$ and increasing $l$ in each instance by 2 (for $d = 5$), 4 (for $d = 4$), or 6 (for $d = 3$) to reach hard instances in all categories.

In Childsnack one has to prepare sandwiches, where some children may eat only certain kind of ingredients (e.g. gluten-free) (Fuentetaja and de la Rosa 2016). The dimensionality parameter $n$ is the number of contents on each sandwich (modeled as a predicate $P(s, c_1, \ldots, c_n)$), which is normally fixed but which we scale here. Each child has preferences, e.g., allowing only tomatoes and salad. We create different variants scaling the number of children (3, 5, and 7), which is also the number of goals. In each category, we scale task size by increasing the amount of contents available, as well as more generous preferences for the children.

Finally, for (c) we include huge instances of IPC Blocksworld, Logistics, and Rovers, keeping the goal simple enough so that some tasks are within reach for current lifted planners (Ridder and Fox (2014) ran a similar experiment, but the benchmarks are not publicly available).

For Blocksworld, we scale the number of blocks from 100 to 1900, increasing by 200 blocks per instance. In the initial state, all blocks are placed on the table (we experimented with arbitrary initial states but were unable to find instances too hard to ground yet within reach of lifted planners). For Logistics, all tasks contain one city, one airplane, one truck, and ten packages. We scale the number of locations starting with 1000 and increasing by 250 in each instance. For Rovers, we generated tasks with a single rover, one objective and one camera. We scale the number of waypoints starting from 1000 and increasing by 500 in each instance.

### Results

Table 1 shows coverage results. L-RPG is not competitive, which must be interpreted with care given the implementation differences. We remark that, as intended in our design, our heuristic functions are very fast. Indeed, the node generation rate (number of generated states per second) is almost up to the standards of goal counting: on average across all benchmark tasks, $h^{\text{gc}}$ is only 1.37 times faster than $h^{\text{ur}}$ (max 3.34) and 1.67 times faster than $h^{\text{ur-d}}$ (max 3.47).

In Organic Synthesis, plans are usually short, and most of the complexity lies in successor generation, to the effect that all PWL configurations have the same coverage. In GED and Pipesworld, our heuristics are not informative. This is partly

**Table 1** layout — columns: max. arity ($\mathcal{P}$, $\mathcal{A}$); #ground; Grounded (FD): $|\mathcal{A}^\mathcal{O}|$, $|\mathcal{P}^\mathcal{O}|$, $h^{\text{gc}}$, $h^{\text{FF}}$; Lifted: L-RPG, BFS, $h^{\text{gc}}$, $h^{\text{ur}}$, $h^{\text{ur-d}}$, $h^{\text{gc,ur}}$, $h^{\text{gc,ur-d}}$.

### (1) Large action schema arity

| | $\mathcal{P}$ | $\mathcal{A}$ | #ground | $|\mathcal{A}^\mathcal{O}|$ | $|\mathcal{P}^\mathcal{O}|$ | $h^{\text{gc}}$ | $h^{\text{FF}}$ | L-RPG | BFS | $h^{\text{gc}}$ | $h^{\text{ur}}$ | $h^{\text{ur-d}}$ | $h^{\text{gc,ur}}$ | $h^{\text{gc,ur-d}}$ |
|---|---|---|---|---|---|---|---|---|---|---|---|---|---|---|
| ged (156) | 2 | 3 | 156 | 32585 | 1206 | **156** | 62 | 43 | 21 | **156** | 25 | 25 | **156** | **156** |
| ged-spl (156) | 2 | 2 | 156 | 4602 | 734 | **156** | 35 | 58 | 18 | **156** | 18 | 18 | **156** | **156** |
| orgsy-alk (18) | 2 | 16 | 15 | 24475 | 74 | **15** | **15** | 14 | 13 | 13 | 13 | 13 | 13 | 13 |
| orgsy-mit (18) | 2 | 31 | 2 | 2946 | 36 | 2 | 2 | 0 | **6** | **6** | **6** | **6** | **6** | **6** |
| orgsy-org (20) | 2 | 31 | 1 | 137784 | 806 | **1** | **1** | 0 | 0 | 0 | 0 | 0 | 0 | 0 |
| pipeswrl (50) | 3 | 12 | 16 | 119907 | 232 | 16 | 13 | 9 | 11 | **21** | 7 | 7 | 11 | 10 |
| **Sum (418)** | | | 346 | | | 346 | 128 | 122 | 69 | **352** | 69 | 69 | 342 | 341 |

### (2a) Large predicate arity: Visitall

| | $\mathcal{P}$ | $\mathcal{A}$ | #ground | $|\mathcal{A}^\mathcal{O}|$ | $|\mathcal{P}^\mathcal{O}|$ | $h^{\text{gc}}$ | $h^{\text{FF}}$ | L-RPG | BFS | $h^{\text{gc}}$ | $h^{\text{ur}}$ | $h^{\text{ur-d}}$ | $h^{\text{gc,ur}}$ | $h^{\text{gc,ur-d}}$ |
|---|---|---|---|---|---|---|---|---|---|---|---|---|---|---|
| 3d-clo-g1 (10) | 3 | 4 | 7 | 140832 | 48386 | 7 | 7 | 2 | 8 | 8 | **10** | **10** | **10** | **10** |
| 3d-clo-g2 (10) | 3 | 4 | 7 | 140832 | 48388 | 7 | 7 | 2 | 2 | 6 | 8 | 8 | **9** | **9** |
| 3d-clo-g3 (10) | 3 | 4 | 7 | 140832 | 48390 | 7 | 7 | 2 | 1 | 7 | 2 | 1 | **9** | **9** |
| 3d-far-g1 (10) | 3 | 4 | 7 | 140832 | 48386 | 7 | 7 | 2 | 0 | 0 | 1 | **10** | 1 | **10** |
| 3d-far-g2 (10) | 3 | 4 | 7 | 140832 | 48388 | **7** | **7** | 2 | 0 | 0 | 2 | 5 | 2 | **7** |
| 3d-far-g3 (10) | 3 | 4 | 7 | 140832 | 48390 | **7** | **7** | 1 | 0 | 0 | 1 | 4 | 2 | 6 |
| 4d-clo-g1 (10) | 4 | 5 | 3 | 122005 | 33143 | 3 | 3 | 1 | 6 | 6 | **10** | **10** | **10** | **10** |
| 4d-clo-g2 (10) | 4 | 5 | 3 | 122005 | 33145 | 3 | 3 | 1 | 3 | 8 | 9 | 5 | **10** | **10** |
| 4d-clo-g3 (10) | 4 | 5 | 3 | 122005 | 33147 | 3 | 3 | 1 | 1 | 5 | 2 | 1 | **7** | 6 |
| 4d-far-g1 (10) | 4 | 5 | 3 | 122005 | 33143 | 3 | 3 | 1 | 0 | 0 | 1 | **10** | 1 | **10** |
| 4d-far-g2 (10) | 4 | 5 | 3 | 122005 | 33145 | 3 | 3 | 1 | 0 | 0 | 2 | 2 | 2 | **7** |
| 4d-far-g3 (10) | 4 | 5 | 3 | 122005 | 33147 | 3 | 3 | 1 | 0 | 0 | 2 | 2 | 2 | **4** |
| 5d-clo-g1 (10) | 5 | 6 | 2 | 175760 | 40546 | 2 | 2 | 0 | 9 | 9 | **10** | **10** | **10** | **10** |
| 5d-clo-g2 (10) | 5 | 6 | 2 | 175760 | 40548 | 2 | 2 | 0 | 2 | 7 | 4 | 2 | **9** | 8 |
| 5d-clo-g3 (10) | 5 | 6 | 2 | 175760 | 40550 | 2 | 2 | 0 | 1 | 8 | 3 | 4 | **10** | 9 |
| 5d-far-g1 (10) | 5 | 6 | 2 | 175760 | 40546 | 2 | 2 | 0 | 0 | 0 | 2 | **10** | 2 | **10** |
| 5d-far-g2 (10) | 5 | 6 | 2 | 175760 | 40548 | 2 | 2 | 0 | 0 | 0 | 2 | 5 | 2 | **6** |
| 5d-far-g3 (10) | 5 | 6 | 2 | 175760 | 40550 | 2 | 2 | 0 | 0 | 0 | 1 | 2 | 2 | **6** |
| **Sum (180)** | | | 72 | | | 72 | 72 | 17 | 33 | 64 | 72 | 101 | 100 | **147** |

### (2b) Large predicate arity: Childsnack

| | $\mathcal{P}$ | $\mathcal{A}$ | #ground | $|\mathcal{A}^\mathcal{O}|$ | $|\mathcal{P}^\mathcal{O}|$ | $h^{\text{gc}}$ | $h^{\text{FF}}$ | L-RPG | BFS | $h^{\text{gc}}$ | $h^{\text{ur}}$ | $h^{\text{ur-d}}$ | $h^{\text{gc,ur}}$ | $h^{\text{gc,ur-d}}$ |
|---|---|---|---|---|---|---|---|---|---|---|---|---|---|---|
| n1-g3 (12) | 2 | 5 | 12 | 513 | 138 | **12** | **12** | – | 2 | **12** | **12** | **12** | **12** | **12** |
| n1-g5 (12) | 2 | 5 | 12 | 1930 | 218 | 5 | **12** | – | 0 | 3 | 4 | 2 | **12** | **12** |
| n1-g7 (12) | 2 | 5 | 12 | 5011 | 298 | 2 | 4 | – | 0 | 2 | 2 | 1 | 6 | **8** |
| n2-g3 (12) | 3 | 6 | 12 | 9758 | 159 | 5 | 7 | – | 1 | 3 | 5 | 5 | 11 | **12** |
| n2-g5 (12) | 3 | 6 | 12 | 74405 | 253 | 2 | 2 | – | 0 | 1 | 2 | 2 | 3 | **12** |
| n2-g7 (12) | 3 | 6 | 11 | 232344 | 332 | 0 | **2** | – | 0 | 0 | 0 | 0 | 1 | 1 |
| n3-g3 (12) | 4 | 8 | 12 | 65520 | 273 | 3 | 5 | – | 0 | 2 | 2 | 2 | 6 | **11** |
| n3-g5 (12) | 4 | 8 | 7 | 221398 | 364 | 0 | 1 | – | 0 | 0 | 0 | 0 | 1 | **6** |
| n3-g7 (12) | 4 | 8 | 3 | 259615 | 363 | 0 | 0 | – | 0 | 0 | 0 | 0 | 0 | 0 |
| n4-g3 (12) | 5 | 10 | 10 | 176161 | 819 | 0 | 4 | – | 0 | 0 | 3 | 3 | 5 | **10** |
| n4-g5 (12) | 5 | 10 | 1 | 376905 | 348 | 0 | 0 | – | 0 | 0 | 0 | 0 | 0 | **2** |
| n4-g7 (12) | 5 | 10 | 0 | – | – | 0 | 0 | – | 0 | 0 | 0 | 0 | 0 | 0 |
| **Sum (144)** | | | 104 | | | 29 | 49 | – | 3 | 23 | 30 | 27 | 57 | **86** |

### (3) Large object universe

| | $\mathcal{P}$ | $\mathcal{A}$ | #ground | $|\mathcal{A}^\mathcal{O}|$ | $|\mathcal{P}^\mathcal{O}|$ | $h^{\text{gc}}$ | $h^{\text{FF}}$ | L-RPG | BFS | $h^{\text{gc}}$ | $h^{\text{ur}}$ | $h^{\text{ur-d}}$ | $h^{\text{gc,ur}}$ | $h^{\text{gc,ur-d}}$ |
|---|---|---|---|---|---|---|---|---|---|---|---|---|---|---|
| blocks-g2 (10) | 2 | 2 | 2 | 100000 | 50602 | 1 | 2 | 0 | 0 | 2 | **4** | **4** | **4** | **4** |
| blocks-g3 (10) | 2 | 2 | 2 | 100000 | 50602 | 0 | 0 | 0 | 0 | 0 | **2** | **2** | **2** | **2** |
| blocks-g4 (10) | 2 | 2 | 2 | 100000 | 50602 | 0 | 1 | 0 | 0 | 0 | 0 | 0 | **2** | **2** |
| blocks-g5 (10) | 2 | 2 | 2 | 100000 | 50602 | **0** | **0** | 0 | 0 | 0 | 0 | 0 | 0 | 0 |
| logist-g1 (10) | 2 | 4 | 2 | 1282377 | 2252 | 2 | 2 | 0 | **5** | **5** | 0 | 0 | 0 | 0 |
| logist-g2 (10) | 2 | 4 | 2 | 1284629 | 3379 | 2 | 2 | 0 | 0 | **5** | 0 | 0 | 0 | 0 |
| logist-g3 (10) | 2 | 4 | 2 | 1286881 | 4506 | 2 | 2 | 0 | 0 | **5** | 0 | 0 | 0 | 0 |
| logist-g4 (10) | 2 | 4 | 2 | 1289133 | 5633 | 2 | 2 | 0 | 0 | **4** | 0 | 0 | 0 | 0 |
| rovers-g2 (10) | 3 | 6 | 2 | 5316 | 2531 | 2 | 2 | 0 | 0 | 3 | **10** | **10** | **10** | **10** |
| rovers-g4 (10) | 3 | 6 | 1 | 4041 | 2006 | 1 | 1 | 0 | 0 | 1 | **10** | 3 | **10** | 3 |
| rovers-g6 (10) | 3 | 6 | 1 | 4983 | 2026 | 1 | 1 | 0 | 0 | 0 | **10** | 2 | **10** | 2 |
| rovers-g8 (10) | 3 | 6 | 1 | 3753 | 2965 | 1 | 1 | 0 | 0 | 0 | **8** | 1 | **8** | 1 |
| **Sum (120)** | | | 23 | | | 14 | 16 | 0 | 5 | 25 | 44 | 22 | **46** | 24 |

Table 1: Coverage results. $|\mathcal{A}^\mathcal{O}|$ and $|\mathcal{P}^\mathcal{O}|$ show average grounding size for those instances that can be grounded (#ground).

due to a general weakness of delete relaxation here ($h^{\text{gc}}$ is better than $h^{\text{FF}}$ in grounded planning), but partly stems from the information loss in unary splitting.

In our new benchmarks, grounding is hardly possible throughout. In Visitall, expectedly $h^{\text{gc}}$ performs reasonably well when goals are close ("clo"), but not when they are far. Our new heuristics all do better, but particularly with disambiguation. Interestingly, the tie-breaking combination with $h^{\text{gc}}$ works best by far, hinting that our heuristics are unstable and profit from the clear progress identified by reaching more goal atoms. The picture in Childsnack is similar, except here there is no "goal location distance" parameter that we could scale, and $h^{\text{gc}}$ is hopeless throughout. Finally, results in the large IPC domains are mixed. In Logistics, our new heuristics are uninformative and fall far behind $h^{\text{gc}}$. In Blocksworld and Rovers we obtain substantially better search information however. In Rovers, $h^{\text{ur}}$ achieves best results, the only case where disambiguation is systematically detrimental.

## Conclusion

Delete-relaxation heuristics are paramount in classical planning, yet take exponential time in the size of the lifted planning task input. To address this, we have introduced additional relaxations to achieve polynomial-time behavior. We focused on a heuristic that is extremely fast to compute on any lifted task, the unary relaxation, which splits all predicates into unary predicates. Our results with this first simple technique are highly promising and already show that the state of the art can be improved.

However, this barely explores the possibilities of our framework. Exciting avenues opened by this research are, for example, larger tractable fragments of predicate splitting, flexible splitting onto arbitrary sets of parameter tuples, clever methods for choosing such sets, etc.

## Acknowledgments

This work was funded by the Deutsche Forschungsgemeinschaft (DFG, German Research Foundation) — Project-ID 232722074 — SFB 1102.

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
