# OpenReview forum: "Polynomial-Time in PDDL Input Size: Making the Delete Relaxation Feasible for Lifted Planning"
_icaps-conference.org/ICAPS/2021/Workshop/HSDIP — HSDIP 2021_

### Official Review · AnonReviewer1 · 2021-05-25

**Confidence:** 5
**Overall Score:** Strong Accept

**Review:**

The paper introduces unary relaxation of classical planning tasks as a way to achieve the computation of a delete-relaxed heuristic in polynomial time in the size of the lifted task. The paper demonstrates how relaxed plans can be found in such a setting and a heuristic analogous to h^max/FF. On the theoretical side, the authors prove that the computation only takes polynomial time (in the size of the lifted task) and that less relaxed versions of the problem are NP-hard. The proofs are sound, well written, and show an interesting result.

On the empirical side, the new heuristics are evaluated on several hard-to-ground domains. Most of them are specifically designed to showcase the new heuristics and feel somewhat artificial (for example, logistics is scaled up but limited to one city, which makes it essentially equivalent to miconic). Most ways of scaling up the domains just increase the amount of noise in the domain in a way that the new heuristics are robust against. While this demonstrates cases where the heuristics have strengths, it doesn't help to show their limitations.

Over all, this is a high-quality contribution that should be accepted.

### Minor comments:
* Corrêa is missing an accent in several places.
* ether -> whether (page 2)
* proof of theorem 4: extra spacing before "24".
* predicates in text ("at", "next") are sometimes set in italics and sometimes in typewriter font.
* pre_st is sometimes used with indices (pre_st(a, i, j)) and sometimes with objects (pre_st(a, x_i, x_j)).

---

### Official Review · AnonReviewer2 · 2021-05-26
**Review of paper "Polynomial-Time in PDDL Input Size: Making the Delete Relaxation Feasible for Lifted Planning"**

**Confidence:** 5
**Overall Score:** Strong Accept

**Review:**

This paper introduces a new method to compute heuristics for lifted planning
that is polynomial in the PDDL size. The method is a relaxation over the
delete-relaxed task. The main idea is to, for every predicate symbol P with an
associated arity k, create k new versions P_1,P_2,..P_k of P, where P_i is the
(relational algebra) projection of P to its ith element. Using this so-called
unary relaxation, we can compute relaxed plans very quickly and use them as a
heuristic estimate. These techniques perform very well in larger planning tasks
compared to (or sometimes combined with) other lifted methods.

This is a clear acceptance. The topic fits the workshop, the paper is very well
written, the contribution is clear, and the results are interesting. I do not
have any complaints about the paper.

However, I have several questions for the open discussion and also a few
suggestions:

- Did you have the time to compare your approach to Corrêa et al. (ICAPS 2021)?

- In the section about the complexity of lifted relaxed planning, you mention
  that there are still some open questions. What are the open questions you have
  in mind?

- Still on the complexity part: delete-relaxed planning on lifted tasks is
  EXPTIME-complete only if you forbid negated preconditions. It is
  NEXPTIME-complete otherwise (Erol et al. 1995). Are you performing this
  restriction in your results? (I was not sure if pre(a) for some action schema
  a would allow a negated predicate such as ~P(X1,...Xn).) In your empirical
  results, you have domains (e.g., organic-synthesis) with negated
  preconditions.

- I am not sure I understood Theorem 1. Where does the constant predicate arity
  come into play in your proof? Perhaps I am missing something, but I believe
  you can even prove that relaxed plan existence in lifted planning is
  EXPTIME-hard if you reduce it from the Datalog query problem instead of the
  conjunctive query problem.  The reduction should be straightforward: One
  action schema per rule with a single effect (rule head) and conjunctive
  precondition (rule body), the initial state equivalent to the initial EDB, and
  goal condition as the conjunction of the atoms in the query. (The "constant
  arity" part should be easily encoded.)  Completeness would follow from the
  reduction used in Helmert (AIJ 2009) to compute the relaxed reachability
  analysis using a Datalog program. (The only issue would be about negative
  preconditions, but since we are talking about constant or bounded arity, you
  can transform the task into some positive normal form in time and space
  polynomial in this arity.)

- What is the explanation for the disambiguation to underperform in the Rovers
  domain? Is it because it adds too much overhead?

- Is your heuristic inadmissible? If it is, it would be a good idea to mention
  that explicitly.

- You mention that you use Yannakakis' algorithm for the successor generation,
  but I imagine you are citing the wrong paper (Experiments section). The paper
  you cited refers to a fixed-parameter tractable algorithm to compute
  conjunctive queries (with inequalities). I believe you want to cite the 1981
  paper by Yannakakis alone (this is what Corrêa et al. (2020)
  implemented). However, if you implemented the Papadimitriou & Yannakakis FPT
  algorithm, I would be very interested to know about the details.


Some typos and minor comments:
- Introduction: Correa et al. -> Corrêa et al.
- Background: and for all $i \in [1,k] s_i$ -> and, for all $i \in [1,k]$, $s_i$
- Background: findind -> finding
- Background: As running example -> As a running example
- Background (and some other parts): The capitalization of Visitall is
  inconsistent. (i.e., Visitall and visitall)
- Experiments: You are using "FD" to refer to Fast Downward without ever
  introducing it. Also, you introduce the abbreviation "GC (FD)" for the ground
  version of goal count, but you use $h^{gc}$ afterward.
- Experiments: "All experimental results and benchmarks are publicly available"
  -> Final stop is missing. Also, is the source code publicly available?
- Experiments: "We include GED here for historical reasons only" -> I found this
  a bit misleading. GED is still hard-to-ground. The "issue" here is that
  Powerlifted only supports the "easiest" versions, which Fast Downward can
  ground. The positional versions of GED are still very challenging for Fast
  Downward, for example.

---

### Author Response · Authors · 2021-06-01
**Author response**

Thank you for your reviews,

We will address your comments for the final version of the paper.


> Did you have the time to compare your approach to Corrêa et al. (ICAPS 2021)?

  No, both works were done in parallel and our experiments were finished before we knew
  about their work. Our heuristic is based on the FF heuristic which we expect to do
  better than h^{add}. However, we still expect our heuristics to be less informative due
  to performing an additional relaxation, so they only will be superior whenever the
  heuristics by Corrêa et al are too time consuming (e.g. if predicate arity is too high).

  We think there are interesting lines of future work that bridge the gap between both
  heuristics, e.g. by finding more refined predicate splitting techniques that can trade
  off between heuristic informativeness and runtime overhead.




> In the section about the complexity of lifted relaxed planning, you mention that there are still some open questions. What are the open questions you have in mind?
> I am not sure I understood Theorem 1. Where does the constant predicate arity come into play in your proof? Perhaps I am missing something, but I believe you can even prove that relaxed plan existence in lifted planning is EXPTIME-hard if you reduce it from the Datalog query problem instead of the conjunctive query problem. The reduction should be straightforward: One action schema per rule with a single effect (rule head) and conjunctive precondition (rule body), the initial state equivalent to the initial EDB, and goal condition as the conjunction of the atoms in the query. (The "constant arity" part should be easily encoded.) Completeness would follow from the reduction used in Helmert (AIJ 2009) to compute the relaxed reachability analysis using a Datalog program. (The only issue would be about negative preconditions, but since we are talking about constant or bounded arity, you can transform the task into some positive normal form in time and space polynomial in this arity.)

  Yes, we do think that perhaps our results could be strengthened. Here, we were mainly
  interested in showing that delete-relaxation is still hard even in the presence of
  bounded predicate arity. We will clarify in the proof that conjunctive queries are hard
  even if predicates have bounded arity.



> Still on the complexity part: delete-relaxed planning on lifted tasks is EXPTIME-complete only if you forbid negated preconditions. It is NEXPTIME-complete otherwise (Erol et al. 1995). Are you performing this restriction in your results? (I was not sure if pre(a) for some action schema a would allow a negated predicate such as ~P(X1,...Xn).) In your empirical results, you have domains (e.g., organic-synthesis) with negated preconditions.

   Yes, we are dealing with STRIPS without negative preconditions. The version of
   PowerLifted that we started to work with didn't support negated preconditions yet,
   except for equality (the = predicate).

   We can deal with Organic Synthesis because all negated atoms in that domain are of the
   form (not (= ?x ?y)). The unary relaxation itself basically ignores the negated
   equality atoms, since they are trivially true in the unary relaxation as long as there
   is more than one object. We actually extended the disambiguation for handling these
   preconditions by forbidding the same object for both parameters, which is
   straightforward and consistent with our definitions in the paper.



> What is the explanation for the disambiguation to underperform in the Rovers domain? Is it because it adds too much overhead?

 Our hypothesis is that disambiguation is being detrimental due to creating a local
     minima that GBFS has trouble escaping, exploring a huge plateau of irrelevant actions
     before exploring the path to the goal. This does not happen on every instance (on
     other instances of the same domain, disambiguation actually reduces node expansions
     without a big overhead on evaluating the heuristic). These local minima seem to be
     related to having two goals of the form (communicated_image_data X high_res) and
     (communicated_image_data X low_res) for the same value of X.


> Is your heuristic inadmissible? If it is, it would be a good idea to mention that explicitly.

   Yes, we extract relaxed plans as the FF heuristic, so we are also inadmissible. We will clarify this in the paper.


> You mention that you use Yannakakis' algorithm for the successor generation, but I imagine you are citing the wrong paper (Experiments section). The paper you cited refers to a fixed-parameter tractable algorithm to compute conjunctive queries (with inequalities). I believe you want to cite the 1981 paper by Yannakakis alone (this is what Corrêa et al. (2020) implemented). However, if you implemented the Papadimitriou & Yannakakis FPT algorithm, I would be very interested to know about the details.

  You are right, we will correct the citation!

---

### Decision · Program_Chairs · 2021-06-10

**Decision:**

Accept

**Comment:**

Both reviewers strongly suggested to accept the paper and saw no major problems with it.